# Time-of-day changes in physician clinical decision making: A retrospective study

**Peter Trinh** [1]*, **Donald R. Hoover** [2], **Frank A. Sonnenberg** [3]

**1** Rutgers Robert Wood Johnson Medical School, Rutgers University, Piscataway, NJ, United States of America, **2** Department of Statistics and Institute for Health, Health Care Policy and Aging Research, Rutgers University, New Brunswick, NJ, United States of America, **3** Department of Medicine, Rutgers Robert Wood Johnson Medical School, New Brunswick, NJ, United States of America

* Peter.trinh1@gmail.com

**Data Availability Statement:** The dataset generated and analyzed during the current study is available in the Mendeley Data repository, http://dx.doi.org/10.17632/v27rr3zpws.1.

## Abstract

### Background

Time of day has been associated with variations in certain clinical practices such as cancer screening rates. In this study, we assessed how more general process measures of physician activity, particularly rates of diagnostic test ordering and diagnostic assessments, might be affected by time of day.

### Methods

We conducted a retrospective chart review of 3,342 appointments by 20 attending physicians at five outpatient clinics, matching appointments by physician and comparing the average diagnostic tests ordered and average diagnoses assessed per appointment in the first hour of the day versus the last hour of the day. Statistical analyses used sign tests, two-sample t-tests, Wilcoxon tests, Kruskal Wallis tests, and multivariate linear regression.

### Results

Examining physicians individually, four and six physicians, respectively, had statistically significant first- versus last-hour differences in the number of diagnostic tests ordered and number of diagnoses assessed per patient visit ($p \leq 0.04$). As a group, 16 of 20 physicians ordered more tests on average in the first versus last hour ($p = 0.012$ for equal chance to order more in each time period). Substantial intra-clinic heterogeneity was found in both outcomes for four of five clinics ($p < 0.01$).

### Conclusions

There is some statistical evidence on an individual and group level to support the presence of time-of-day effects on the number of diagnostic tests ordered per patient visit. These findings suggest that time of day may be a factor influencing fundamental physician behavior and processes. Notably, many physicians exhibited significant variation in the primary outcomes compared to same-specialty peers. Additional work is necessary to clarify temporal and inter-physician variation in the outcomes of interest.

**Funding:** This study is supported, in part, by Grant Number 1UL1TR003017-01 from the National Center for Advancing Translational Science (https://ncats.nih.gov/). The author associated with this Grant is FAS. The funder had no role in study design, data collection and analysis, decision to publish, or preparation of the manuscript.

**Competing interests:** The authors have declared that no competing interests exist.

# Introduction

Decision-making is fundamental to patient care, and patients and fellow healthcare professionals expect physicians to make clinical decisions in a consistent and deliberative evidence-based manner. However, a growing body of evidence suggests that physicians are not as consistent as expected and may be susceptible to decision fatigue, which is the depletion of self-control and the reduced ability to make decisions and regulate behavior as a result of frequent and recurrent acts of decision making [1, 2].

The study of decision fatigue is derived from psychology studies assessing the nature of self-control. A prominent model in the field is the Strength Model of Self-Control, which posits that self-control originates from an unknown internal resource that functions similarly to a muscle in that it fatigues over time, particularly due to sequential tasks that require self-control [3]. When this internal resource is exhausted or fatigued, one becomes ego-depleted, which is a "state of diminished [mental] resources following exertion of self-control" [4]. In this state, one is more likely to perform worse on subsequent tasks requiring self-control, a phenomenon known as the *sequential task paradigm*. Another model, the mechanistic Process Model of Ego Depletion posits that ego depletion is less the result of the exhaustion of some internal resource and more the product of subconscious shifts in motivation and attention over time that detract from one's ability to regulate impulses and exercise self-control [5]. While the two models each have their merits, decreased cognitive performance over time is a central feature of both.

Decreased cognitive performance over time is particularly pertinent in the context of decision making and decision fatigue. Frequent and recurrent decisions over time have been shown to impair the ability to exercise self-control and make appropriate decisions in ensuing tasks. For instance, in a randomized controlled trial in which the degree of pain tolerance functioned as a proxy for self-control, Vohs and colleagues found that shoppers who had to make many shopping decisions at a grocery store subsequently demonstrated decreased pain tolerance compared to shoppers who only thought about the grocery choices without making decisions [2]. A well-known 2011 study by Danziger et al. examined 1000+ parole decisions and found that the percentage of favorable decisions for defendants gradually decreased from ~65% to near zero as decision making sessions progressed over the course of a day, and this pattern reset and repeated after meal breaks [6]. Both studies arguably demonstrate a temporal effect with decision making and decision fatigue.

In medicine, several studies have shown temporal variations in clinical outcomes that are explained potentially by decision fatigue. For example, in one study by Persson et al., the probability of orthopedic surgeons deciding to operate on a patient was strongly associated with a patient's appointment time, with probabilities steadily declining throughout the day [7]. Additionally, rates of influenza vaccination and clinician ordering of cancer screening tests have been shown to significantly decline over the course of a day [8, 9], and similar patterns were observed in studies examining primary care providers' antibiotic and opioid prescribing practices as well as hand-washing compliance in hospitals [10–12]. These findings clearly suggest that physicians' ability to make rational, evidence-based patient care decisions may suffer as a day goes on, and patients seen later in a clinic day may experience suboptimal care compared to those seen at the start of the day. Such temporal variations in clinical outcomes have significant implications for healthcare quality.

To our knowledge, studies examining temporal variations and the potential influence of decision fatigue on clinical decision making have been limited to assessing specific clinical decisions. Therefore, we aimed to investigate variations in more general process measures of clinical activity that could potentially reflect fundamental physician behavior on a more generalizable dimension and spur greater thought and action to mitigate variation in healthcare quality.

We examined variations in two general process measures: the number of diagnostic tests ordered and the number of diagnoses assessed per patient encounter. We are not aware of any prior studies that have utilized these outcomes in the assessment of temporal variation and decision fatigue. Both were chosen because unlike outcomes in prior studies, these are generalizable to most physicians; almost all clinicians must go through the decision-making process of ordering tests and assessing diagnoses in an electronic medical record (EMR) system in order to properly care for patients. Also, given that those experiencing decision fatigue tend to exhibit avoidant behaviors like procrastination, deferment, or complete avoidance of decisions during subsequent decision making [1, 2, 7, 13], temporal variations in the number of these decisions could be a suitable proxy for decision fatigue.

Moreover, these process measures can be used as markers of healthcare quality. Diagnostic test ordering has important implications for the general practice of medicine, healthcare costs, and clinical outcomes. Estimates of the rates of unnecessary tests vary from 10–50% of all orders [14, 15], and such inappropriate testing can have significant downstream effects on patients as a result of follow-up tests, prolonged hospital stays, patient dissatisfaction, and unnecessary referrals or procedures [16, 17]. Alternatively, a landmark study showed that patients seen by their physicians receive only approximately 55% of recommended care, including diagnostic tests, for preventive, acute, and chronic conditions [18]. In considering the number of diagnoses assessed per patient appointment, research has suggested that quality of care typically rises as a patient's number of medical conditions rises [19]. Assuming that the number of diagnoses assessed by a physician rises with the number of medical conditions a patient has, the number of diagnoses may be an indicator of healthcare quality, possibly as a reflection of the thoroughness of a clinical encounter. How thoroughly clinicians evaluate their patients may vary due to time of day and decision fatigue.

Based on the behavioral theories of ego depletion and evidence that both repeated decision-making leads to decision fatigue and that those who are depleted are more likely to exhibit avoidant behavior like deferring decisions [2, 7], we conducted a retrospective analysis of patient records to assess the outcomes of the number of diagnostics tests ordered per patient appointment and the number of diagnoses assessed per patient appointment. We hypothesized that physicians ordered fewer diagnostic tests and assessed fewer diagnoses per patient appointment for patients seen during the last hour versus the first hour of the day.

## Methods

This study was approved by the institutional review board of New Brunswick Health Sciences, Rutgers, the State University of New Jersey. A waiver of informed consent was granted due to the study's minimal risk and infeasibility of informed consent given the study's retrospective design.

### Setting and participants

Patient data from 5,354 unique appointments by 39 attending physicians from January 1 to December 31, 2017 were gathered retrospectively from the AthenaFlow EMR (formerly known as GE Centricity) used by outpatient clinics in the Department of Medicine of Rutgers Robert Wood Johnson Medical School. The study authors captured all encounters from six different specialty clinics in order to control for inter-specialty differences in diagnostic test ordering patterns. The cardiology, endocrinology, hematology, nephrology, general internal medicine, and rheumatology practices clinics were chosen because they tend to differ in the number and type of diagnoses managed and tests ordered. To preserve provider confidentiality, clinic identities were masked during data extraction and analysis, and physicians were designated by numbers.

Outpatient clinic sessions generally occur in two 4-hour sessions, typically either morning (8AM-12PM) or afternoon (1PM-5PM). Some physicians see outpatients only in the morning or only in the afternoon, and some see patients for a full day. Providers and the associated patient visit data were included in the study if 1) the providers held full clinic days (both morning and afternoon), since prior studies have demonstrated significant temporal differences in decision outcomes between the start and end of days [6–8], 2) the clinicians had at least 25 patient-visits in the first hour (~8-9AM) and at least 25 patient-visits in the last hour (~4-5PM) of the above-described days, cumulative over the entire year, to ensure adequate sample size, and 3) the providers were attending-level physicians. Patient visits were excluded if patients were seen by residents and fellows and if patients were younger than 18. Using these criteria, data from 3,342 unique patient appointments by 20 attending physicians at five clinics were included; no doctors from Clinic Two were included due to lack of physicians who saw patients for a full day.

## Data collection

Patient encounters were matched by physician, and to maximize potential for differential cumulative work fatigue, only data from the morning's first hour (~8-9AM) and the afternoon's last hour (~4-5PM) were analyzed. The start time of an encounter was defined as the EMR time stamp on the first entry in the History of Present Illness section in the EMR. Appointments were counted as first-hour appointments if the start time was within one hour of the start of the first appointment of the day. For example, if the first appointment of the day started at 8:00am, then an appointment that started at 8:59am was counted as occurring in the first hour, while an appointment starting at or after 9:00am was not. Appointments were identified as last-hour appointments if, according to the clinic schedule, they ended less than one hour prior to the end of the last appointment of the afternoon. For instance, if the last appointment ended at 5:15pm, an appointment that started at 3:50pm and ended at 4:20pm was categorized as being in the last hour, but one that started at 3:50pm and ended at or before 4:15pm was not.

The number of diagnoses assessed during each encounter was determined by the number of ICD-10 codes associated with the Evaluation and Management (E&M) billing code for the encounter. In the AthenaFlow EMR, all diagnoses recorded in the Assessment and Plan portion of EMR notes are automatically associated with the E&M code. Additional data collected included patient age, sex, race, ethnicity, health insurance, number of diagnostic tests ordered, number of diagnoses evaluated, and patient primary and secondary diagnoses via ICD-10 codes. Patients' active problems were collected to calculate the Charlson comorbidity index [20].

## Statistical analysis

ANOVA models with physician as a covariate assessed the overall differences (first and last hour combined) in means between physicians at the same clinic sites, and two sample t-tests assessed the differences in first- versus last-hour means for the same physician for the primary outcomes. As the distributions of numbers of diagnoses made and diagnostic tests ordered per visit were often skewed, nonparametric Kruskal Wallis and Wilcoxon tests, respectively, were also used to obtain *p*-values for the same-clinic site physician differences and the same-physician first versus last hour differences. To confirm that statistically significant Wilcoxon and Kruskal Wallis *p*-values seen in unadjusted analyses remained statistically significant in case mix-adjusted analyses ($p < 0.05$), to the best of our ability to do so, multivariate linear regression was used. Linear models of the primary outcomes comparing i) all physicians at the same

site, and ii) within each individual physician, first versus last hour, were fit and adjusted for patient age, sex, race, ethnicity, health insurance, and Charlson comorbidity index. No confirmatory case mix-adjusted comparisons were made for physicians with nonsignificant Wilcoxon and Kruskal Wallis $p$-values. For the number of diagnostic tests ordered per appointment, the multivariate models also adjusted for number of diagnoses made per appointment, and the overall comparisons of physicians within clinics were adjusted for first versus last hour.

It should be noted that as a result of i) our belief that first versus last hour differences varied from physician to physician, ii) skewness of the data and limited numbers of observations for some physicians, and iii) other violations of needed assumptions, such as homogeneity of variance between physicians, we did not find it feasible nor informative to fit more complicated, repeated measures linear models that pooled all physicians together. However, we did use the sign test to compare numbers of physicians that had more diagnoses made (or more laboratory tests ordered) in the first versus last hour to see if a directional, across-physician trend existed. All $p$-values reported are two-sided. Analyses were conducted using SAS Version 9.4 (Cary, NC), and statistical tests considered $p < 0.05$ to be statistically significant.

## Results

The study included 20 physicians in five practices with 3,342 total patient appointments: 2,013 in the first hour and 1,329 in the last hour. Patient characteristics, except perhaps for sex, were similar between patients seen in the first versus the last hour of the day (Table 1). About 55% of first-hour appointments and 61% of last-hour appointments were with female patients, but this difference varies by physician; the Breslow-Day test of homogeneity for equality of proportion of female patients in the first versus last hour between physicians is 0.025. Fourteen physicians had a greater proportion of female patients in last-hour visits. Eighteen physicians had more patient encounters in the first hour of the day versus the last hour compared to only one clinician with the opposite trend (sign test $p < 0.001$) (Table 2). A single physician had the same number of appointments in each time period.

### Primary outcomes for individual physicians

Tables 2 and 3 display results of the primary outcomes when comparing each individual physician to themselves at the beginning versus the end of the day. Medians for both primary outcomes for each individual clinician is provided in S1 and S2 Tables.

Table 2 displays the within-physician time-of-day differences in the mean numbers of diagnostic tests ordered per patient appointment. For example, Physician 2 in Clinic 1 ordered an average of 3.3 ± 2.13 diagnostic tests per patient visit in the first hour compared to 2.95 ± 2.2 in the last hour, corresponding to a mean difference of -0.35 tests per patient visit ($p = 0.58$ for equality by the Wilcoxon test). Overall, for this outcome, sixteen physicians had no statistically significant differences between the first and last hour. Only four physicians had both statistically significant, unadjusted (Wilcoxon) and adjusted multivariate linear regression differences in diagnostic tests ordered per patient encounter between the first and last hours of their day. Of these four, three had ordered more diagnostic tests per encounter on average in the first hour compared to the last hour of the workday (adjusted $p$-values ranging from 0.012 to < 0.001). Conversely, Physician 37 had fewer mean tests per encounter in the first hour (14.86 tests vs. 21.10 tests, adjusted $p < 0.001$).

Table 3 displays similar data for the number of diagnoses assessed per appointment. For instance, Physician 2 in Clinic 1 assessed 3.12 ± 1.32 diagnoses per patient visit in the first hour compared to 2.66 ± 1.19 in the last hour, corresponding to a mean difference of -0.46

**Table 1. Sample demographics of patients.**

| | No. (%) | |
|---|---|---|
| Characteristic | First Hour | Last Hour |
| Patients, No. | 2013 | 1329 |
| Age, mean (SD), years | 56.4 (16.6) | 56.3 (17.7) |
| Gender | | |
| Male | 908 (45.1) | 515 (38.8) |
| Female | 1104 (54.8) | 814 (61.2) |
| Unspecified | 1 (0.0) | 0 (0.0) |
| Ethnicity | | |
| Hispanic or Latino | 317 (15.7) | 192 (14.4) |
| Not Hispanic or Latino | 1663 (82.6) | 1120 (84.3) |
| Unspecified | 33 (1.6) | 17 (1.3) |
| Race | | |
| American Indian or Alaska Native | 4 (0.2) | 7 (0.5) |
| Asian | 176 (8.7) | 138 (10.4) |
| Black or African American | 373 (18.5) | 276 (20.8) |
| Native Hawaiian or Other Pacific Islander | 6 (0.3) | 6 (0.5) |
| White | 1102 (54.7) | 685 (51.5) |
| Unspecified | 352 (17.5) | 217 (16.3) |
| Insurance | | |
| Private | 1224 (60.8) | 770 (57.9) |
| Medicare | 735 (36.5) | 509 (38.3) |
| Medicaid | 6 (0.3) | 2 (0.2) |
| Not Recorded | 48 (2.4) | 48 (3.6) |
| Charlson Comorbidity Index | | |
| 0 | 679 (33.7) | 394 (29.6) |
| 1 | 535 (26.6) | 386 (29.0) |
| 2+ | 799 (39.7) | 549 (41.3) |

diagnoses per patient visit ($p = 0.006$ for equality by the Wilcoxon test). Because this $p$-value was $\leq 0.05$, a confirmatory case mix-adjusted comparison was made, yielding a $p$-value of 0.01. Overall, for this outcome, 14 physicians had no statistically significant unadjusted or adjusted differences between the first and last hour of their day, but six physicians did. Five of these six physicians made more unadjusted and adjusted mean diagnoses per encounter in the first hour of their day compared to the last hour (adjusted $p$-values ranging from 0.02 to $<0.001$), while there was again one physician with the opposite pattern (Physician 35: 3.55 diagnoses assessed in the first hour vs. 3.86 in the last hour, adjusted $p < 0.04$).

## Assessing for a collective temporal trend in the primary outcomes

Despite detecting only a handful of doctors with statistically significant time-of-day differences in the primary outcomes, these findings do not statistically rule out time-of-day differences for other physicians. Thus, we were interested in assessing post hoc if there was a statistically significant group-level trend toward more tests ordered or more diagnoses assessed in the first versus the last hour of the day. We found that 80% of the clinicians (16 of the 20) ordered more diagnostic tests per appointment on average in the first hour of the day compared to 20% (4 out of 20) who ordered more in the last hour of the day (two-sided $p = 0.012$ by exact test for each physician to have equal probability to order more during each time period). For

**Table 2. Differences in mean number of laboratory tests ordered.**

| Clinic Location | Provider | Number of Appointments | | Means ± Std-Dev | | | P Values | |
|---|---|---|---|---|---|---|---|---|
| | | First Hour | Last Hour | First Hour | Last Hour | Difference | Wilcoxon | P-Genmod |
| One | 2 | 27 | 19 | 3.3 ± 2.13 | 2.95 ± 2.2 | -0.35 | 0.58 | ÷ |
| | 6 | 33 | 21 | 0.52 ± 0.83 | 0.33 ± 0.91 | -0.19 | 0.18 | ÷ |
| | 7 | 143 | 80 | 1.05 ± 1.38 | 0.81 ± 1.23 | -0.24 | 0.18 | ÷ |
| | 8 | 21 | 16 | 0.38 ± 0.97 | 0.75 ± 1.48 | 0.37 | 0.55 | ÷ |
| | 10 | 294 | 172 | 0.59 ± 1.15 | 0.53 ± 1.02 | -0.06 | 0.98 | ÷ |
| Three | 17 | 29 | 29 | 4.59 ± 4.57 | 4.48 ± 4.01 | -0.11 | 0.87 | ÷ |
| | 18 | 43 | 22 | 8.44 ± 2.6 | 7.95 ± 2.42 | -0.49 | 0.74 | ÷ |
| Four | 20 | 27 | 19 | 7.81 ± 6.14 | 6.96 ± 5.73 | -0.85 | 0.46 | ÷ |
| | 22 | 48 | 27 | 5.39 ± 2.91 | 1.63 ± 2.16 | -3.76 | <0.001* | <0.001* |
| Five | 25 | 197 | 126 | 3.07 ± 2.58 | 1.83 ± 2.46 | -1.24 | <0.001* | 0.012* |
| | 26 | 18 | 16 | 1.67 ± 2.2 | 1.81 ± 2.43 | 0.14 | 0.74 | ÷ |
| | 27 | 21 | 31 | 1.48 ± 2.04 | 1.29 ± 2.05 | -0.17 | 0.74 | ÷ |
| | 29 | 32 | 19 | 3.09 ± 3.33 | 2.32 ± 2.5 | -0.77 | 0.56 | ÷ |
| | 30 | 290 | 215 | 2.91 ± 3.14 | 1.84 ± 2.31 | -1.07 | 0.0002* | 0.002* |
| | 31 | 61 | 29 | 2.28 ± 2.41 | 2.14 ± 2.66 | -0.14 | 0.58 | ÷ |
| | 32 | 64 | 36 | 2.52 ± 2.29 | 1.08 ± 1.73 | -1.44 | <0.001* | 0.24 |
| Six | 34 | 188 | 130 | 8.15 ± 8.18 | 8.4 ± 9.22 | 0.35 | 0.68 | ÷ |
| | 35 | 213 | 132 | 5.45 ± 4.05 | 5.2 ± 3.61 | -0.25 | 0.9 | ÷ |
| | 36 | 109 | 83 | 11.88 ± 9.25 | 11.82 ± 8.79 | -0.06 | 0.98 | ÷ |
| | 37 | 141 | 93 | 14.96 ± 9.54 | 21.81 ± 10.76 | 6.85 | <0.001* | <0.001* |

÷ Confirmatory adjusted *p*-values not taken due to low sample size and/or non-statistically significant unadjusted p-values.

** Statistically significant *p*-values.

the diagnoses assessed outcome, 60% of doctors (12 out of 20) assessed more diagnoses on average per patient encounter in the first hour compared to 40% (8 out of 20) who assessed more diagnoses on average in the last hour (two-sided *p* = 0.50 by exact test).

## Same-specialty variation in the primary outcomes

Finally, while not part of the original study objective, we noticed considerable variation in the primary outcome data from physician to physician, prompting us to assess how physicians performed relative to their same-specialty peers. Tables 4 and 5 display the aggregate mean and median data for the primary outcomes for all physicians. Physicians in Clinics One, Three, Five, and Six all had statistically significant within-clinic differences in their practice patterns for each primary outcome compared to their same-specialty peers (*p*-values for equality ranging from 0.01 to <0.001). An evident example in Clinic Six is Physician 35 who had a mean of 3.88 lab tests ordered per appointment while his or her peer, Physician 37, had a much larger mean of 17.68 tests per appointment (Table 4). Only Clinic Four had statistically nonsignificant within-clinic physician differences for both outcomes, except in the unadjusted analysis of diagnostic test orders.

## Discussion

Among a group of 20 outpatient physicians, there is statistical evidence to support the existence of time-of-day effects on diagnostic test ordering and diagnostic assessments, the study's proxies for physician decision making. Statistically significant time-of-day differences in the

**Table 3. Differences in mean number of diagnoses assessed.**

| Clinic Location | Provider | Number of Appointments | | Means ± Std-Dev | | | P Values | |
|---|---|---|---|---|---|---|---|---|
| | | First Hour | Last Hour | First Hour | Last Hour | Difference | Wilcoxon | P-Genmod |
| One | 2 | 27 | 19 | 6.37 ± 1.9 | 6.11 ± 2.11 | -0.26 | 0.78 | ÷ |
| | 6 | 33 | 21 | 2.12 ± 0.78 | 1.71 ± 0.84 | -0.41 | 0.06 | ÷ |
| | 7 | 143 | 80 | 3.12 ± 1.32 | 2.66 ± 1.19 | -0.46 | 0.006* | 0.01* |
| | 8 | 21 | 16 | 1.29 ± 0.46 | 1.38 ± 0.72 | 0.09 | 1 | ÷ |
| | 10 | 294 | 172 | 1.94 ± 0.94 | 1.92 ± 0.87 | -0.02 | 0.19 | ÷ |
| Three | 17 | 29 | 29 | 3.76 ± 0.51 | 3.9 ± 0.41 | 0.14 | 0.33 | ÷ |
| | 18 | 43 | 22 | 4.95 ± 1.56 | 5.41 ± 1.76 | 0.46 | 0.27 | ÷ |
| Four | 20 | 27 | 19 | 2.73 ± 1.3 | 3.11 ± 1.12 | 0.38 | 0.13 | ÷ |
| | 22 | 48 | 27 | 2.85 ± 1.44 | 2.73 ± 1.44 | -0.12 | 0.69 | ÷ |
| Five | 25 | 197 | 126 | 3.93 ± 1.65 | 2.95 ± 1.65 | -0.98 | <0.001* | <0.001* |
| | 26 | 18 | 16 | 3.61 ± 2.5 | 2.12 ± 1.89 | -1.49 | 0.05* | 0.02* |
| | 27 | 21 | 31 | 2.67 ± 1.11 | 2.39 ± 0.95 | -0.28 | 0.48 | ÷ |
| | 29 | 32 | 19 | 3.19 ± 1.4 | 3.74 ± 1.88 | 0.55 | 0.45 | ÷ |
| | 30 | 290 | 215 | 2.83 ± 1.39 | 2.28 ± 1.34 | -0.55 | <0.001* | <0.001* |
| | 31 | 61 | 29 | 3.16 ± 1.69 | 3.17 ± 1.83 | 0.01 | 0.92 | ÷ |
| | 32 | 64 | 36 | 3.98 ± 1.54 | 2.78 ± 1.57 | -1.2 | <0.001* | 0.007* |
| Six | 34 | 188 | 130 | 2.89 ± 1.67 | 2.78 ± 1.02 | -0.11 | 0.34 | ÷ |
| | 35 | 213 | 132 | 3.55 ± 1.21 | 3.86 ± 1.22 | 0.31 | 0.004* | 0.04* |
| | 36 | 109 | 83 | 2.94 ± 1.09 | 2.87 ± 1.12 | -0.07 | 0.5 | ÷ |
| | 37 | 141 | 93 | 4.59 ± 1.69 | 4.68 ± 1.91 | 0.09 | 0.52 | ÷ |

÷ Confirmatory adjusted *p*-values not taken due to low sample size and/or non-statistically significant unadjusted *p*-values.

** Statistically significant *p*-values.

average number of tests ordered and diagnoses assessed per patient visit were found in a non-negligible minority of physicians, and the directional trend at the group level for diagnostic test orders support our hypothesis that clinicians would order more tests per patient in the first hour compared to the last hour of the day. Interestingly, there was also substantial variation in the primary outcomes between physicians of the same specialty. Altogether, these findings demonstrate the need for further investigation.

If time of day affects decision making related to test orders and diagnostic assessment for at least some doctors, we believe that based on observations from prior clinical studies [7–11], decision fatigue and ego depletion may be a mediating factor. Tasks like ordering lab tests and assessing diagnoses require executive cognitive function, and as decision fatigue progresses as a day goes on, physicians may become ego-depleted and subconsciously exhibit avoidant behavior by forgoing additional tasks and decisions [2, 7]. This leads to fewer actions performed per patient appointment. The statistically significant, within-physician level data and the group level data is largely consistent with this hypothesis. For each outcome, all but one of the minority of physicians with statistically significant differences ordered more tests or assessed more diagnoses per patient in the first hour compared to the last hour of the day. The group level data also demonstrated that most clinicians ordered more diagnostics tests per patient encounter at the start versus the end of the day.

The fact that most individual physicians had statistically nonsignificant time-of-day differences can be interpreted in several ways. First, from a behavioral standpoint, it could suggest

**Table 4. Overall number of laboratory tests ordered by providers in first and last hour combined.**

| Clinic Location[a] | Provider | Total Appointments (First + Last Hour) | Mean Tests Ordered Per Appointment ± Std-Dev | Median (95% CI) | Q3 (95% CI) | Kruskal Wallis p-value[b] | P-Genmod p-value[b] |
|---|---|---|---|---|---|---|---|
| One | 2 | 46 | 3.15 ± 2.14 | 3 (2,4) | 5 (4,6) | <0.001 | <0.001 |
| | 6 | 54 | 0.44 ± 0.86 | 0 (0,0) | 1 (0,2) | | |
| | 7 | 223 | 0.96 ± 1.33 | 0 (0,1) | 1 (1,2) | | |
| | 8 | 37 | 0.54 ± 1.22 | 0 (0,0) | 0 (0,2) | | |
| | 10 | 466 | 0.57 ± 1.10 | 0 (0,0) | 1 (1,1) | | |
| Three | 17 | 58 | 4.53 ± 4.25 | 4 (2,5) | 7 (5,10) | <0.001 | <0.001 |
| | 18 | 65 | 8.28 ± 2.53 | 8 (8,9) | 10 (9,11) | | |
| Four | 20 | 75 | 5.97 ± 1.94 | 7 (5,8) | 10 (8,12) | <0.001 | 0.22 |
| | 22 | 74 | 3.72 ± 3.19 | 4 (2,5) | 6 (6,7) | | |
| Five | 25 | 323 | 2.59 ± 2.60 | 2 (2,2) | 4 (4,5) | 0.006 | 0.01 |
| | 26 | 34 | 1.74 ± 2.27 | 1 (0,2) | 3 (1,5) | | |
| | 27 | 52 | 1.38 ± 2.03 | 0 (0,1) | 3 (1,4) | | |
| | 29 | 51 | 2.80 ± 3.05 | 1 (1,4) | 6 (3,6) | | |
| | 30 | 505 | 2.46 ± 2.86 | 1 (1,2) | 4 (3,5) | | |
| | 31 | 90 | 2.33 ± 2.48 | 1.5 (1,2) | 4 (3,5) | | |
| | 32 | 100 | 2.21 ± 1.20 | 1 (1,2) | 2 (3,4) | | |
| Six | 34 | 318 | 8.25 ± 1.98 | 5 (4,7) | 11 (10,14) | <0.001 | <0.001 |
| | 35 | 345 | 3.88 ± 0.64 | 6 (5,6) | 8 (7,9) | | |
| | 36 | 192 | 11.85 ± 9.03 | 10 (8,12) | 18 (16,21) | | |
| | 37 | 234 | 17.68 ± 10.57 | 16 (15,18) | 25 (23,28) | | |

[a] Each Clinic Location represents a specific specialty.

[b] For within-clinic equality of providers.

that those doctors with no significant differences consistently engaged in behavior that mitigated time-of-day effects and ego depletion. Studies have shown that ego depletion of self-control can interestingly be countered by personal factors such as beliefs about self-control, moods, and self-affirmations [21–23]. It is entirely plausible that most study physicians, who by default are an accomplished group of individuals, held positive self-affirming thoughts about their abilities and self-control, whereas a few doctors lacked those perspectives, leaving them particularly susceptible to time-of-day effects. Alternatively, it is possible that the use of EMR decision support tools, which aim to reduce the number of decisions and actions for physicians, varied amongst study physicians to the point that those who did not use such tools were more susceptible to temporal effects. Indeed, a study by Kim et al. demonstrated a substantial role for EMR decision support in reducing same-day, temporal disparities in influenza vaccination rates [9].

Second, from a statistical perspective, the lack of statistically significant temporal differences for the majority of study physicians does not necessarily mean that such differences do not exist for those clinicians. Instead, this result could be due to Type II error, as many of the clinicians had small numbers of patient visits in the data set. Despite the inability to pinpoint with statistical certainty a temporal difference in primary outcomes for every individual doctor, the collective finding that i) some individual physicians had significant temporal differences, and ii) there was a significant group-level temporal trend for diagnostic tests ordered, is considerable statistical evidence for an association between time of day and decision making. While the group-level trend was statistically significant only for the diagnostic test outcome, this underscores that we cannot rule out potential time-of-day effects on physician decision making.

**Table 5. Overall number of diagnoses assessed by providers in first and last hour combined.**

| Clinic Location[a] | Provider | Total Appointments (First + Last Hour) | Mean Diagnoses Assessed Per Appointment ± Std-Dev | Median (95% CI) | Q3 (95% CI) | Kruskal Wallis p-value[b] | P-Genmod p-value[b] |
|---|---|---|---|---|---|---|---|
| One | 2 | 46 | 6.26 ± 2.02 | 5 (8,17) | 8 (8,29) | <0.001 | <0.001 |
| | 6 | 54 | 1.96 ± 0.75 | 2 (2,2) | 2 (2,3) | | |
| | 7 | 223 | 2.96 ± 1.29 | 3 (3,3) | 4 (4,4) | | |
| | 8 | 37 | 1.32 ± 0.58 | 1 (1,1) | 2 (1,2) | | |
| | 10 | 466 | 1.90 ± 0.91 | 2 (2,2) | 2 (2,3) | | |
| Three | 17 | 58 | 3.83 ± 0.46 | 4 (4,4) | 4 (4,4) | <0.001 | <0.001 |
| | 18 | 65 | 5.11 ± 1.63 | 5 (5,6) | 6 (6,7) | | |
| Four | 20 | 75 | 2.87 ± 1.24 | 3 (2,3) | 4 (3,4) | 0.49 | 0.66 |
| | 22 | 74 | 2.80 ± 1.43 | 3 (2,3) | 3 (3,4) | | |
| Five | 25 | 323 | 3.55 ± 1.69 | 4 (3,4) | 5 (3,5) | <0.001 | <0.001 |
| | 26 | 34 | 2.91 ± 2.33 | 2 (1,4) | 4 (3,7) | | |
| | 27 | 52 | 2.50 ± 1.02 | 3 (2,3) | 3 (3,3) | | |
| | 29 | 51 | 3.39 ± 1.60 | 3 (3,4) | 4 (4,5) | | |
| | 30 | 505 | 2.59 ± 1.39 | 2 (2,3) | 4 (3,4) | | |
| | 31 | 90 | 3.17 ± 1.72 | 3 (2,4) | 4 (4,5) | | |
| | 32 | 100 | 3.55 ± 1.65 | 4 (3,4) | 4.5 (4,5) | | |
| Six | 34 | 318 | 2.84 ± 1.05 | 3 (3,3) | 4 (3,4) | <0.001 | <0.001 |
| | 35 | 345 | 3.67 ± 1.22 | 4 (4,4) | 4 (4,4) | | |
| | 36 | 192 | 2.90 ± 0.78 | 3 (3,3) | 3.5 (3,4) | | |
| | 37 | 234 | 4.62 ± 1.78 | 5 (4,5) | 6 (5,6) | | |

[a] Each Clinic Location represents a specific specialty.

[b] For within-clinic equality of providers.

Alternative explanations for time-of-day, decision fatigue-mediated associations include the tendency for doctors to fall behind schedule as the clinic day progresses, compressing the time available for assessment and task completion for each last-hour patient. This idea holds less weight considering that collectively there were fewer visits in the last hour (1329 visits) compared to the first hour (2013 visits), and only one doctor saw more patients in the last hour compared to the first hour. Another alternative is that both patients and clinicians may experience a desire to leave sooner at the end of the day [8], leaving less time for patients to participate in further evaluation and for clinicians to perform additional tasks. Lastly, purposeful scheduling of more or less complex patients for certain parts of the day is a possibility, although the temporal differences for the minority of physicians persisted after case-mix adjustment.

It is important to note that in both subsets of physicians with statistically significant differences, one physician exhibited contrasting behavior by ordering more tests or assessing more diagnoses per patient on average in the last hour of the day. This result may support an alternative or complementary hypothesis on the effect of decision fatigue on decision making–that sometimes given an ego-depleted state, individuals may act more impulsively [24]. When physicians generate a diagnostic workup plan for patient care, an initial step is to think of what diagnostics tests to order. To hone this plan, a second cognitive step involves assessing if each test is truly necessary. This second step is arguably often skipped or forgotten, as evidenced by the large estimated rates of unnecessary tests [14, 15]. From an ego-depletion perspective, Physician 37's end-of-day behavior of ordering more tests per patient on average may represent either an impulsive tendency to order more tests than necessary as a form of defensive medicine, or an avoidance of the follow-up cognitive step to evaluate test necessity. For Physician

35 who assessed more diagnoses per patient at the end of the day, their behavior could possibly represent impulsive entry of unconfirmed differential diagnoses into the EMR or procrastination in the removal of disproved diagnoses from the EMR.

While the underlying causative factors are unclear, time-of-day differences in the primary outcomes, and by extension physician decision making, would suggest that quality of clinical care varies by time of day. Such variation carries important implications for healthcare operations and practices. However, this study was unable to determine how quality of care was temporally affected since defining and assessing quality of care, including whether study physicians were over or undertesting or over or underassessing patients by time of day, was beyond the scope of the study data.

Interestingly, unanticipated nontemporal variation was observed between same-specialty physicians with respect to the primary outcomes. Clinicians at four clinic locations had considerable case mix-adjusted differences among themselves and their same-specialty peers in both the numbers of diagnostic tests ordered and the number of diagnoses assessed per patient appointment. While these differences likely reflect factors such as age, experience, within-specialty expertise around complex disorders, and different backgrounds in residency or fellowship training [25], they do imply variation in quality of care delivered within singular outpatient clinics. Whether one doctor delivers better or worse care compared to his or her same-specialty peer at the same clinic location is unclear, but such variation alone merits further investigation.

This work has several important limitations. First, the study is observational, and consequently, the results are subject to unmeasured confounders, one of which is true duration of appointment. Because there was no reliable data point in the EMR system that accurately reflected actual appointment end times, we made the assumption that the duration of the appointment was the scheduled duration. Additionally, literature has suggested that the greater the difficulty of the decision, the more decision fatigue an individual may face [26]. We attempted to control for decision complexity using the Charlson comorbidity index as a proxy, but this may be an imperfect measure. Second, because the initial EMR data extraction was limited to a one-year period and many physicians at the study site do not often work full outpatient days in the same clinic, this comparison of first versus last hour of a full day was significantly underpowered for many of the original 39 physicians, resulting in fewer data points and limited generalizability of the study. The single-site design of the study also contributed to its limited generalizability. Additionally, no demographic data was collected on providers per IRB concerns. While such data could have provided key insights into same-specialty physician differences observed, possible unmeasured confounders of first versus last hour within-physician differences, and physician susceptibility to decision fatigue, the provider population at the study institution is sufficiently small that collecting and reporting clinician demographic data could jeopardize provider confidentiality. Importantly, this study did not attempt to measure physician decision fatigue directly, and as a result, relationships between the outcomes of interest, decision fatigue, and time of day are limited to inferences. Lastly, some of the within-physician mean time-of-day differences, while statistically significant, are small enough in magnitude that they may not be clinically significant for individual patients. It should be noted, however, that small differences can aggregate into large differences over time and patient visit volume, which from a health systems perspective, may shed valuable light on the quality of care delivered by physicians as a whole.

## Conclusion

There is some statistical evidence on an individual and group level to support the existence of time-of-day effects on clinician decision making, particularly on the number of diagnostic

tests ordered per patient. These findings suggest that time of day may be a factor influencing fundamental physician behavior and processes. Notably, many physicians also exhibited significant variation in the primary outcomes compared to same-specialty peers. Additional work is necessary to clarify time-of-day effects and inter-physician variation in the outcomes of interest.

## Supporting information

**S1 Table. Differences in median number of laboratory tests ordered.**
(TIF)

**S2 Table. Differences in median number of diagnoses assessed.**
(TIF)

## Acknowledgments

Contributors: The authors would like to acknowledge John Francis for his key contribution to the acquisition of data for the study.

## Author Contributions

**Conceptualization:** Peter Trinh.

**Data curation:** Frank A. Sonnenberg.

**Formal analysis:** Donald R. Hoover.

**Funding acquisition:** Frank A. Sonnenberg.

**Investigation:** Peter Trinh, Frank A. Sonnenberg.

**Methodology:** Peter Trinh, Donald R. Hoover, Frank A. Sonnenberg.

**Project administration:** Peter Trinh.

**Resources:** Frank A. Sonnenberg.

**Software:** Donald R. Hoover.

**Supervision:** Peter Trinh, Frank A. Sonnenberg.

**Validation:** Donald R. Hoover.

**Visualization:** Peter Trinh.

**Writing – original draft:** Peter Trinh, Donald R. Hoover, Frank A. Sonnenberg.

**Writing – review & editing:** Peter Trinh, Donald R. Hoover, Frank A. Sonnenberg.

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
