## [Decision Letter · Decision Letter 0]

26 May 2021

PONE-D-21-13464

Time-of-Day Changes in Physician Clinical Decision Making: A Retrospective Study

PLOS ONE

Dear Dr. Trinh,

Thank you for submitting your manuscript to PLOS ONE. After careful consideration, we feel that it has merit but does not fully meet PLOS ONE’s publication criteria as it currently stands. Therefore, we invite you to submit a revised version of the manuscript that addresses the points raised during the review process.

We look forward to receiving your revised manuscript.

Kind regards,

Paola Iannello

Academic Editor

PLOS ONE

Journal Requirements:

Reviewers' comments:

Reviewer's Responses to Questions

**Comments to the Author**

1. Is the manuscript technically sound, and do the data support the conclusions?

Reviewer #1: No

Reviewer #2: Partly

2. Has the statistical analysis been performed appropriately and rigorously? 

Reviewer #1: Yes

Reviewer #2: Yes

3. Have the authors made all data underlying the findings in their manuscript fully available?

Reviewer #1: No

Reviewer #2: Yes

4. Is the manuscript presented in an intelligible fashion and written in standard English?

Reviewer #1: Yes

Reviewer #2: Yes

5. Review Comments to the Author

Reviewer #1: The paper deals with the decision fatigue in the medical decision making context. The authors discussed the time-of-day changes in daily physicians’ decision making. They compared the impact of work fatigue between first hour in the morning and last hour in the afternoon on number of diagnostic tests ordered and number of diagnosis assessed per patient appointment.

One main concern regarding the paper is related to the fact that a theoretical background is completely missing. One theory that may be useful could be the Strength Model of Self-Control (Baumeister et al., 1998); another one the Process Model of Ego Depletion (Inzlicht, Schmeichel, 2012). I suggest the authors to better frame their introduction with reference to the most update theories on decision fatigue.

A definition of the decision fatigue has been given, but I guess in a wrong way. The authors use Hsiang et al., 2019 as reference paper, but Hsiang et al quote Vohs et al 2008 when they give the definition of decision fatigue.

(Vohs KD, Baumeister RF, Schmeichel BJ, Twenge JM, Nelson NM, Tice DM. Making choices impairs subsequent self-control: a limited-resource account of decision making, self-regulation, and active initiative. J Pers Soc Psychol. 2008;94(5):883-898. doi:10.1037/0022-3514.94.5.883)

There are several variables that the authors did not consider in their study. For example, they did not control for the complexity or difficult of decisions. Literature* suggests that the higher is the difficulty rises by a decision, the more decision fatigue an individual experiences. In their study, we do not know anything about the king of decisions the physicians took and we cannot be sure the decisions can be compared each other.

*Oto, B (2012) When thinking is hard: Managing decision fatigue. EMS World 41(5): 46–50.

The authors should better clarify in the introduction what is the expected direction in the relationship between time of day and decision fatigue.

One of the most important flaw of the paper is that it does not measure the decision fatigue itself. The time-of-day during which the decisions are taken is used as proxy of decision fatigue, but the authors did not measure the decision fatigue levels of the doctors.

The fact that the time-of-day influences the decision fatigue is already know in literature. What is the novelty of this study?

Moreover, there are studies in literature in which results demonstrated that who is experience decision fatigue may be either passive/avoidant or impulsive. Then, in some cases it seems that decision fatigue acts increasing procrastination, passive behavior, low persistence, and the choice of a default option; whereas, in others, individuals can act impulsively. All of this may impact on the medical decision making either in the way the authors hypothesized (low number of diagnostic tests ordered and diagnosis assessed or in the opposite way.

Regarding the method, I have a question for the authors: how can we know that number of diagnostic tests ordered and number of diagnosis assessed were lower at the end of the day just because those patients needed less tests than the patients visited in the early morning? Another thing is: is there any way to know the physicians’ characteristics? We do not know if they have different characteristics that may explain the results. A similar consideration can be made for the patients.

I have one question again for the authors regarding the results: in Table 2 and 3, they reported the number of appointments divided for first hour and last hour. It seems to me that, for example, in clinic number 1, the provider number 2 had 27 appointments in the first hour and 19 in the last hour. Perhaps I do not understand very well this point, but it seems quite odd.

The tables are cut on their right sides.

Thank you very much for the opportunity to read and revise this paper.

Reviewer #2: Thank you for giving me the opportunity to review the paper entitled “Time-of-day changes in physician clinical decision making: a retrospective study”, submitted to PlosOne. I think this manuscript addresses a paramount topic and I agree with the importance of analyze factors affecting clinical decision and in particular decision fatigue in clinical context. However, I would like to suggest some changes that I hope the authors will consider.

Introduction

The introduction is clear and there is a concise description of the concept of the measures used within the project. However, I don’t understand some fundamental points

• there is no/little reference framework, which is supported in the literature, on the use of these measures to identify the “decision fatigue” in physician clinical decision making context. Are there references in the literature that justify the choice of these two measurement indices (the number of diagnostic tests ordered and the number of diagnoses assessed during a clinical encounter)? "The study of variations in more general measures of clinical activity, such as frequency of diagnostic test ordering, could potentially reflect physician behavior on a wider dimension, including capturing the impact of fatigue on decision making." (line 83-86). Please specify the references or argue more about the choice of indicators.

• It may be useful to broaden the analysis of the literature on the importance of time-of-day in clinical practice (see Shuchman, M. (2019). Does time of day matter in clinical practice?).

• For the purpose of understanding and linearity of reading, I would consider it useful to add in the final part of the introduction the hypotheses of research and the questions to be answered in an orderly manner in the section dedicated to results and discussion.

• the "primary outcomes" paraghaph is not very clear, in relation to its position in the Manuscript text. Is it possible to move it and integrate it with the hypothesis, at the end of the introduction?

Setting and Participants

Regarding clinic identities (line 122-127), is it possible to have information about the association between the number with which the clinic is identified in the study and the specialty? While respecting the privacy and anonymity of participants and preserving provider confidentiality, it would be useful to know the association between the clinic and their specific field of work.

Please specify more clearly and in a more structured way which were the criteria for inclusion and exclusion in the study design. Furthermore, it is not clear which inclusion/exclusion criteria have not been met by physicians from “Clinic 2”, which has not been considered in the results sections.

I wonder whether the authors could be a little more specific in what they considered as “Cumulative Work Fatigue” and “decision fatigue” and whether they can discuss (in the Introduction and/or Discussion section) their study (and their results) in light with literature works on repeated decision making process and associated decision fatigue. The analysis of the pros and cons in the decision-making process requires cognitive commitment; this is one of the reasons why, when you are more tired, you tend to avoid a reasoning that requires cognitive commitment (See for example Persson, E., Barrafrem, K., Meunier, A., & Tinghög, G. (2019). The effect of decision fatigue on Surgeons' clinical decision making. Health economics, 28(10), 1194-1203; Vohs, K. D., Baumeister, R. F., Schmeichel, B. J., Twenge, J. M., Nelson, N. M., & Tice, D. M. (2008). Making Choices Impairs Subsequent Self-control: A Limited-Resource Account of Decision Making, Self-regulation, and Active Initiative. Journal of Personality and Social Psychology, 94(5), 883-898. https://doi.org/10.1037/0022-3514.94.5.883.).

Furthermore, it would be useful to understand why you choose to analyze the first and last hour of the day in order to maximize potential for differential cumulative work fatigue. With respect to work fatigue, the time of the day between 1pm to 3 pm are not taken into account, unless someone consider them to be the time-of-day in which major at work-accidents occur precisely because of fatigue and sleep. What studies have you referred to considering the first and last hour of the day as being more impacted by fatigue at work?

Results

In Table 2 and Table 3 I suggest to put a star next to the statistically significant values in the table, to make them quickly identifiable

Discussion

Line 300-301: Can you give examples of behaviors that could mitigated time-of-day effects?

Line 303-304: Can you suggest some explanations to why some doctors might be more susceptible to the time-of-day effect on their clinical performance than others? Could they be more susceptible to cognitive bias? Could depend on personal characteristics such as seniority of service or personality characteristics? Please report some references to your suggestion or hypothesis.

Line 330-336: It is interesting to note that in the discussion is inserted and contemplated the possibility of a different/inverse effect of fatigue on the decision making process. Are there any references/other studies that have detected this cognitive effect? Please argue.

Conclusions

I think conclusions should be more cautious given the different limitations identified in the study.

6. PLOS authors have the option to publish the peer review history of their article (what does this mean?). If published, this will include your full peer review and any attached files.

Reviewer #1: No

Reviewer #2: **Yes: **Giulia Ongaro

---

## [Author Response · Author response to Decision Letter 0]

5 Jul 2021

Paola Iannello

Academic Editor

PLOS ONE

Dear Dr. Iannello and Reviewers,

On behalf of my co-authors, I’d like to thank you very much for your thoughtful comments about our manuscript. Incorporating your constructive feedback has enabled us to produce a more robust manuscript. Below is a point-by-point response to each of the reviewers. The reviewer comments are in quotations, and our response follows each quotation block. I hope that we have addressed and clarified each point fittingly, and we look forward to your response.

Sincerely,

Peter Trinh, MD, MBA

Peter.trinh1@gmail.com

+1-973-747-5664

Reviewer #1:

“One main concern regarding the paper is related to the fact that a theoretical background is completely missing. One theory that may be useful could be the Strength Model of Self-Control (Baumeister et al., 1998); another one the Process Model of Ego Depletion (Inzlicht, Schmeichel, 2012). I suggest the authors to better frame their introduction with reference to the most update theories on decision fatigue.”

 Thank you for this crucial feedback. We fully acknowledge this shortcoming and have updated the Introduction to incorporate a theoretical foundation. Please see Page 3, Lines 81-120.

“A definition of decision fatigue has been given, but I guess in a wrong way. The authors use Hsiang et al., 2019 as reference paper, but Hsiang et al quote Vohs et al 2008 when they give the definition of decision fatigue.

(Vohs KD, Baumeister RF, Schmeichel BJ, Twenge JM, Nelson NM, Tice DM. Making choices impairs subsequent self-control: a limited-resource account of decision making, self-regulation, and active initiative. J Pers Soc Psychol. 2008;94(5):883-898. doi:10.1037/0022-3514.94.5.883)”

Thank you. We have updated the manuscript with the appropriate citation.

“There are several variables that the authors did not consider in their study. For example, they did not control for the complexity or difficult of decisions. Literature* suggests that the higher is the difficulty rises by a decision, the more decision fatigue an individual experiences. In their study, we do not know anything about the king of decisions the physicians took and we cannot be sure the decisions can be compared each other.

*Oto, B (2012) When thinking is hard: Managing decision fatigue. EMS World 41(5): 46–50.”

The reviewer makes a great point about the possible effects of the complexity/difficulty of decisions. Making a choice about whether to order a diagnostic test or assess a diagnosis is ultimately a binary decision. In the context of clinical practice, what makes these binary decisions more difficult is arguably the patient’s medical complexity, which is represented in this study by the Charlson comorbidity index (CCI) (Reference #20). We believe that by controlling for this index, similar to other studies (Reference #8-9), and by comparing early encounters in a given specialty only with late encounters in the same specialty, we have, by proxy, controlled for the difficulty of the decisions. According to Table 1, there were no remarkable differences in CCI between patients in the first and last hours. Please see Page 21, Lines 606-609 for our in-text edits.

“The authors should better clarify in the introduction what is the expected direction in the relationship between time of day and decision fatigue.”

To address this point, we have edited the Introduction to delineate our hypothesis more clearly. Please see Page 6, Lines 206-212.

“One of the most important flaws of the paper is that it does not measure the decision fatigue itself. The time-of-day during which the decisions are taken is used as proxy of decision fatigue, but the authors did not measure the decision fatigue levels of the doctors.”

 Thank you for this feedback. This is indeed a critical point, and we have edited the limitations section of the Discussion to acknowledge this. Please see Page 22, Line 575. This is a potential subject for future study.

“The fact that the time-of-day influences the decision fatigue is already know in literature. What is the novelty of this study?”

 The novelty of the study is to expand the literature on the temporal effect of decision fatigue in medicine beyond the very clinically specific outcomes that presently exist in the literature. Studies in the medical field have only looked at specific decisions/outcomes that are particular to their medical specialty. Please see References #7-11. Our outcomes (diagnostic tests ordered and diagnoses assessed per patient) are much more general process measures that can be applied to many more medical specialties. We have edited the Introduction to hopefully delineate this point more clearly to readers. Please see Page 4, Lines 133-162.

“Moreover, there are studies in literature in which results demonstrated that who is experience decision fatigue may be either passive/avoidant or impulsive. Then, in some cases it seems that decision fatigue acts increasing procrastination, passive behavior, low persistence, and the choice of a default option; whereas, in others, individuals can act impulsively. All of this may impact on the medical decision making either in the way the authors hypothesized (low number of diagnostic tests ordered and diagnosis assessed or in the opposite way.”

This is an excellent point. We have integrated these findings from the literature into the Introduction and our interpretation of the results in the Discussion section to address this point of feedback. Please see Page 5, Line 169 (Introduction), Page 18, Line 457 and Page 20, Lines 519-529 (Discussion).

“Regarding the method, I have a question for the authors: how can we know that number of diagnostic tests ordered and number of diagnosis assessed were lower at the end of the day just because those patients needed less tests than the patients visited in the early morning? Another thing is: is there any way to know the physicians’ characteristics? We do not know if they have different characteristics that may explain the results. A similar consideration can be made for the patients.”

This is a great question. To try and address this, we used the Charlson comorbidity index (CCI), which is a measure of medical complexity of patients (Ref #20). According to our analysis, patients in the morning versus the afternoon seemed to be no different from each other, especially in the CCI score, so it’s less likely that patients at the end of the day had less tests just because those patients needed less tests. This is based on the assumption that patients with similar CCI scores would likely have similar numbers of diagnostic tests ordered and numbers of diagnoses assessed. As with all things, we unfortunately cannot definitively rule out that there were other patient differences as well.

With regard to the physicians, we stratified the analyses comparing the first to the last hour within the same physician, so in effect, our comparisons were within-physician (i.e. what he or she did in the afternoon compared to what he/she did in the morning). Since the characteristics of the physician remain the same in both time periods, these cancel out in the comparison. We would have liked to study further these associations for interactions with physician characteristics but were unable to collect this data. Due to the small size of the study site, the IRB felt it best to mask physician demographics and which specialties were associated with each clinic in order to protect provider confidentiality. We acknowledge how provider characteristics could certainly influence these results and have edited language in the limitation section of the Discussion section (Page 22, Lines 570-575) to communicate this message more effectively.

“I have one question again for the authors regarding the results: in Table 2 and 3, they reported the number of appointments divided for first hour and last hour. It seems to me that, for example, in clinic number 1, the provider number 2 had 27 appointments in the first hour and 19 in the last hour. Perhaps I do not understand very well this point, but it seems quite odd.”

We are not sure why the number of appointments in the last hour of the day was smaller than in the first hour, but it was a consistent empirical finding in the outpatient practices we studied. In fact, of 20 providers, 18 had more appointments in the first hour than the last hour compared to one physician who had the same number of first and last hour appointments (p < 0.0001). This difference may relate to scheduling strategies (e.g. scheduling new or extended appointments near the end of the day) or to a greater tendency for appointments later in the day to be cancelled. We would also like to note that despite the number of appointments in the first and last hours not being the same for each physician, our outcomes of interest were purposely “per appointment” outcomes and not “per hour” outcomes that could have been influenced by the number of patient appointments the physicians had.

“The tables are cut on their right sides.”

Thank you for catching this. The Tables have been edited to address this issue.

Reviewer #2:

“The introduction is clear and there is a concise description of the concept of the measures used within the project. However, I don’t understand some fundamental points.

• there is no/little reference framework, which is supported in the literature, on the use of these measures to identify the “decision fatigue” in physician clinical decision making context. Are there references in the literature that justify the choice of these two measurement indices (the number of diagnostic tests ordered and the number of diagnoses assessed during a clinical encounter)? "The study of variations in more general measures of clinical activity, such as frequency of diagnostic test ordering, could potentially reflect physician behavior on a wider dimension, including capturing the impact of fatigue on decision making." (line 83-86). Please specify the references or argue more about the choice of indicators.”

Thank you very much for this crucial feedback. We have edited the Introduction to incorporate a theoretical foundation and more thorough reference framework behind decision fatigue. Please see Page 3, Lines 81-120. 

In our literature search, we could not find any studies that utilized the number of diagnostic tests ordered and the number of diagnoses assessed to examine time-of-day effects, and one of the reasons we chose these parameters was due to their wide applicability within medicine. Please see Page 4, Line 133-205 for our edited Introduction that reflects more clearly our rationale for choosing these parameters. 

“• It may be useful to broaden the analysis of the literature on the importance of time-of-day in clinical practice (see Shuchman, M. (2019). Does time of day matter in clinical practice?).”

In conducting our literature research, we were able to find only a handful of studies that analyzed the importance of time-of-day in clinical practice. We have cited these studies in our Introduction (Page 4, Lines 121-128). Coincidentally, these studies are the same studies referred to by M. Shuchman in her 2019 review article. Please see References #8-11.

“• For the purpose of understanding and linearity of reading, I would consider it useful to add in the final part of the introduction the hypotheses of research and the questions to be answered in an orderly manner in the section dedicated to results and discussion.”

Thank you for this feedback. We have edited the Introduction as the reviewer suggested to communicate our hypothesis more clearly. Please see Page 6, Lines 206-212.

“• The "primary outcomes" paragraph is not very clear, in relation to its position in the Manuscript text. Is it possible to move it and integrate it with the hypothesis, at the end of the introduction?”

Thank you. We agree with this feedback and have edited the Introduction and Methods to address this feedback and improve the flow of the paper. Please see Page 6, Lines 206-212.

“Setting and Participants

Regarding clinic identities (line 122-127), is it possible to have information about the association between the number with which the clinic is identified in the study and the specialty? While respecting the privacy and anonymity of participants and preserving provider confidentiality, it would be useful to know the association between the clinic and their specific field of work.”

Due to the size of the study site, the Institutional Review Board (IRB) felt it best to mask both physician demographics and which specialties were associated with each clinic in order to best protect provider confidentiality. We thus did not collect the physician characteristics. We made sure to acknowledge how provider characteristics could certainly have influenced these results in the limitations paragraph of our Discussion (Page 21, Lines 570-575). In a future large-scale study, we hope to capture provider characteristics, including specialty, without compromising confidentiality.

“Please specify more clearly and in a more structured way which were the criteria for inclusion and exclusion in the study design. Furthermore, it is not clear which inclusion/exclusion criteria have not been met by physicians from “Clinic 2”, which has not been considered in the results sections.”

We have edited the Methods section to address this feedback and convey the exclusion/inclusion criteria more clearly. Please see Page 7, Lines 240-250.

I wonder whether the authors could be a little more specific in what they considered as “Cumulative Work Fatigue” and “decision fatigue” and whether they can discuss (in the Introduction and/or Discussion section) their study (and their results) in light with literature works on repeated decision making process and associated decision fatigue. The analysis of the pros and cons in the decision-making process requires cognitive commitment; this is one of the reasons why, when you are more tired, you tend to avoid a reasoning that requires cognitive commitment (See for example Persson, E., Barrafrem, K., Meunier, A., & Tinghög, G. (2019). The effect of decision fatigue on Surgeons' clinical decision making. Health economics, 28(10), 1194-1203; Vohs, K. D., Baumeister, R. F., Schmeichel, B. J., Twenge, J. M., Nelson, N. M., & Tice, D. M. (2008). Making Choices Impairs Subsequent Self-control: A Limited-Resource Account of Decision Making, Self-regulation, and Active Initiative. Journal of Personality and Social Psychology, 94(5), 883-898. https://doi.org/10.1037/0022-3514.94.5.883.).

We very much appreciate the reviewer’s feedback on this point and the provided citations. To address this, we have modified the Introduction to incorporate a theoretical foundation (Lines 81-120) and reference prior studies looking at decision fatigue within medicine (Lines 121-128). We have also edited the Discussion to discuss our study in the context of the literature and theory on ego depletion and decision fatigue (Page 18, Lines 453-474; Page 20, 519-529).

“Furthermore, it would be useful to understand why you chose to analyze the first and last hour of the day in order to maximize potential for differential cumulative work fatigue. With respect to work fatigue, the time of the day between 1pm to 3 pm are not taken into account, unless someone consider them to be the time-of-day in which major at work-accidents occur precisely because of fatigue and sleep. What studies have you referred to considering the first and last hour of the day as being more impacted by fatigue at work?”

Our idea was to maximize the potential to detect same-day differences by taking and comparing the two extremes. One extreme where the physician is assumedly fresh and has just started the workday, and the other extreme being a timepoint in which the physician has accumulated the most amount of decision fatigue or is the most ego-depleted due to the accumulated amount of work and decisions they’ve made over the course of the day (as one would infer according to the sequential task paradigm). This study design was, in part, inspired by the results of the following studies (Ref #6-8): 

Danziger S, Levav J, Avnaim-Pesso L. Extraneous factors in judicial decisions. Proc Natl Acad Sci U S A [Internet]. 2011 Apr 26 [cited 2018 Apr 16];108(17):6889–92. Available from: http://www.ncbi.nlm.nih.gov/pubmed/21482790

Persson E, Barrafrem K, Meunier A, Tinghög G. The effect of decision fatigue on surgeons’ clinical decision making. Heal Econ (United Kingdom) [Internet]. 2019 Oct 1 [cited 2021 Jun 10];28(10):1194–203. Available from: /pmc/articles/PMC6851887/

Hsiang EY, Mehta SJ, Small DS, Rareshide CAL, Snider CK, Day SC, et al. Association of Primary Care Clinic Appointment Time With Clinician Ordering and Patient Completion of Breast and Colorectal Cancer Screening. JAMA Netw Open [Internet]. 2019 May 10 [cited 2019 May 20];2(5):e193403. Available from: http://jamanetworkopen.jamanetwork.com/article.aspx?doi=10.1001/jamanetworkopen.2019.3403

Each of these studies tracked their respective outcomes by hour over the course of a day, and each demonstrated significant differences in their outcomes between the first and last hours of the day. With consistent first versus last hour differences found in these prior studies, we thought it sufficient to simplify our analysis to the first versus last hour of a day as well.

“Results

In Table 2 and Table 3 I suggest to put a star next to the statistically significant values in the table, to make them quickly identifiable.”

We have modified the Tables with an asterisk to demarcate the statistically significant values.

“Discussion

Line 300-301: Can you give examples of behaviors that could mitigated time-of-day effects?”

To address this point, we have edited the Discussion (please see Page 18, Line 468-470) and referenced appropriate studies that demonstrate examples of mitigating factors to time-of-day effects/ego depletion, such as personal beliefs about self-control, self-affirmations, and emotional mood. The studies referenced are below (Ref #21-23):

Schmeichel BJ, Vohs KD. Self-Affirmation and Self-Control: Affirming Core Values Counteracts Ego Depletion. J Pers Soc Psychol [Internet]. 2009 Apr [cited 2021 Jun 19];96(4):770–82. Available from: https://pubmed.ncbi.nlm.nih.gov/19309201/

Tice DM, Baumeister RF, Shmueli D, Muraven M. Restoring the self: Positive affect helps improve self-regulation following ego depletion. J Exp Soc Psychol. 2007 May 1;43(3):379–84. 

Job V, Dweck CS, Walton GM. Ego depletion-is it all in your head? implicit theories about willpower affect self-regulation. Psychol Sci [Internet]. 2010 Nov [cited 2021 Jun 19];21(11):1686–93. Available from: https://pubmed.ncbi.nlm.nih.gov/20876879/

“Line 303-304: Can you suggest some explanations to why some doctors might be more susceptible to the time-of-day effect on their clinical performance than others? Could they be more susceptible to cognitive bias? Could depend on personal characteristics such as seniority of service or personality characteristics? Please report some references to your suggestion or hypothesis.”

 Susceptibility and mitigation of time-of-day effects could be two sides of the same coin. We referenced the aforementioned studies above that demonstrated how mood, self-affirmation, and views on self-control could mitigate ego depletion and decision fatigue. Individuals who are more susceptible to time-of-day effects could simply be ones who do not exhibit these mitigating behaviors. Alternatively, it’s possible that physicians have variable use of decision support tools in the electronic medical record system such that those who are not using decision support tools may be more susceptible to time-of-day effects and decision fatigue. Studying the use of decision support and its effect on decision is fatigue is a topic of interest to us for further study. We have edited the Discussion to incorporate these possible explanations (please see Page 18, Line 466-479).

“Line 330-336: It is interesting to note that in the discussion is inserted and contemplated the possibility of a different/inverse effect of fatigue on the decision making process. Are there any references/other studies that have detected this cognitive effect? Please argue.”

Thank you for this feedback. We have edited the manuscript and referenced a 2011 article by Tierney (Ref #34), who has cowritten books with Dr. Baumeister, to better ground our discussion with the extant literature. This article explains the possibilities of impulsive versus avoidant behaviors that may result from decision fatigue and thus could possibly explain the relative outlier physicians in our study. Please see Page 20, Lines 515-529.

“Conclusions

I think conclusions should be more cautious given the different limitations identified in the study.”

 We agree that we should be cautious and conservative with our conclusions. We have further hedged the conclusions by editing it to “some statistical evidence.” Overall, we tried to be as conservative as possible with our study conclusions, using less firm language such as “some” “suggest” and “may.”

---

## [Decision Letter · Decision Letter 1]

3 Sep 2021

Time-of-Day Changes in Physician Clinical Decision Making: A Retrospective Study

PONE-D-21-13464R1

Dear Dr. Trinh,

We’re pleased to inform you that your manuscript has been judged scientifically suitable for publication and will be formally accepted for publication once it meets all outstanding technical requirements.

Kind regards,

Paola Iannello

Academic Editor

PLOS ONE

Additional Editor Comments (optional):

Reviewers' comments:

Reviewer's Responses to Questions

**Comments to the Author**

1. If the authors have adequately addressed your comments raised in a previous round of review and you feel that this manuscript is now acceptable for publication, you may indicate that here to bypass the “Comments to the Author” section, enter your conflict of interest statement in the “Confidential to Editor” section, and submit your "Accept" recommendation.

Reviewer #1: All comments have been addressed

Reviewer #2: All comments have been addressed

2. Is the manuscript technically sound, and do the data support the conclusions?

Reviewer #1: Yes

Reviewer #2: Yes

3. Has the statistical analysis been performed appropriately and rigorously? 

Reviewer #1: Yes

Reviewer #2: Yes

4. Have the authors made all data underlying the findings in their manuscript fully available?

Reviewer #1: Yes

Reviewer #2: Yes

5. Is the manuscript presented in an intelligible fashion and written in standard English?

Reviewer #1: Yes

Reviewer #2: Yes

6. Review Comments to the Author

Reviewer #1: The authors have answered to all my questions. Thank you for their work, I think the paper is now publishable

Reviewer #2: I thank the author very much for their thoughtful response to this round of reviews. In my opinion, this revised version was much improved. I suggest only authors to pay more attention to the responses to the reviewers in particular with reference to the indications for lines and pages that are not at any point correct. This makes it more difficult to identify corrections and changes made to the Manuscript.

7. PLOS authors have the option to publish the peer review history of their article (what does this mean?). If published, this will include your full peer review and any attached files.

Reviewer #1: No

Reviewer #2: No

---

## [Editor Report · Acceptance letter]

8 Sep 2021

PONE-D-21-13464R1 

Time-of-day changes in physician clinical decision making: a retrospective study 

Dear Dr. Trinh:

I'm pleased to inform you that your manuscript has been deemed suitable for publication in PLOS ONE. Congratulations! Your manuscript is now with our production department. 

Kind regards, 

on behalf of

Dr. Paola Iannello 

Academic Editor

PLOS ONE